

# Real-time discrimination of photon pairs using machine learning at the LHC

**Sean Benson[1], Adrián Casais Vidal[2], Xabier Cid Vidal[2] and Albert Puig Navarro[3⋆]**

**1** Nikhef National Institute for Subatomic Physics, Amsterdam, The Netherlands
**2** Instituto Galego de Física de Altas Enerxías (IGFAE),
Universidade de Santiago de Compostela, Santiago de Compostela, Spain
**3** Physik-Institut, Universität Zürich, Zürich, Switzerland

## Abstract

ALP–mediated decays and other as-yet unobserved $B$ decays to di-photon final states are a challenge to select in hadron collider environments due to the large backgrounds that come directly from the $pp$ collision. We present the strategy implemented by the LHCb experiment in 2018 to efficiently select such photon pairs. A fast neural network topology, implemented in the LHCb real-time selection framework achieves high efficiency across a mass range of 4–20 GeV/$c^2$. We discuss implications and future prospects for the LHCb experiment.

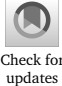
# 1 Introduction

The $\gamma\gamma$ final state is interesting for a variety of reasons. On one hand, the decay of a $B_s^0$ or a $B^0$ meson to two photons remains unobserved and is described by an annihilation topology. It is sensitive to contributions from physics beyond the Standard Model of Particle Physics (SM) including a fourth generation [1], an extended Higgs sector [2] and SUSY [3]. Previous measurements by the Belle and BaBar collaborations have set limits of $\mathcal{B}(B_s^0 \to \gamma\gamma) < 3.1 \times 10^{-6}$ at 90 % confidence level (CL) [4] and $\mathcal{B}(B^0 \to \gamma\gamma) < 3.3 \times 10^{-7}$ at 90 % CL [5], which are significantly above the SM predictions of $\mathcal{B}(B_s^0 \to \gamma\gamma) \sim (2-37) \times 10^{-7}$ and $\mathcal{B}(B^0 \to \gamma\gamma) \sim (1-10) \times 10^{-8}$ [6].

Undiscovered particles known as Axion-Like Particles (ALPs) could also be accessed through this final state. ALPs are pseudo Nambu-Goldstone bosons, associated to spontaneously broken approximate symmetries, which appear in several models and can solve many of the SM problems [7]. Probing very small couplings of ALPs to the SM sets indirect constraints on the New Physics (NP) scale. For the type of model addressed in Ref. [7], ALPs couple to gluons (which allows their production at the LHC) or photons (which can be used for their detection) in the SM sector. The mass of ALPs can be arbitrarily below the NP scale. In particular, for ALPs with a mass in the range between 5 and 10 GeV/$c^2$, the LHCb experiment, described in Ref. [8], has unique sensitivity for their discovery provided they can be selected by its trigger algorithms [7]. For masses below 5 GeV/$c^2$, LHCb may have sensitivity through other decay channels, as described in Ref. [9].

The maximum rate at which events can be read out of the LHCb detector is imposed by the front-end electronics and corresponds to $\sim 1$ MHz. In order to determine which events are kept, hardware triggers based on field-programmable gate arrays are used with a fixed latency of 4 µs. Information from the ECAL, HCAL, and muon stations is used in separate hardware-level (L0) triggers. As explained in Ref. [10], the strategy at L0 for the photon pairs under study here is similar to that of other radiative decays and relies on the Photon or Electron channels, based on inputs from the ECAL. All events selected by L0 are transferred to the High Level Trigger (HLT). The HLT is a software application, executed on an event filter farm, that is implemented in the same Gaudi framework [11] as the software used for the offline reconstruction. It consists of two levels: an initial selection of high energy and/or displaced single- or double-particle signatures (HLT1) and a second level (HLT2), in which full offline reconstruction is available, allowing for more complex searches to be performed [12].

The study of these purely neutral modes at LHCb is challenging, but the use of $\gamma \to e^+ e^-$ conversions in the detector material, which happens for around 25 % of photons, provides additional information to reduce the background levels. With offline selections already in place since Run 1, this paper describes the trigger strategy adopted in Run 2, where a set of trigger selections were introduced to select the $\gamma\gamma$ signature for the case of zero, one, and two photon conversions, labelled as *0CV*, *1CV* and *2CV*, respectively. An additional label *LL* and *DD* is used to distinguish photon conversions reconstructed as:

- Long tracks (*LL*): when all possible tracking information from every tracking station is available, implying that the parent particle decayed within about 1 metre of the $pp$ interaction point

- Downstream tracks (*DD*): using only information from tracking stations different to the vertex locator, implying that the parent particle decayed after this.

The work presented in this paper builds on the strategy first introduced in 2015 to select only the $B_{(s)}^0$ decay [10], in which a selection was put in place for *0CV*, *1CV LL*, *1CV DD*, and *2CV* (which includes both *LL* and *DD* combinations). The new approaches to select candidates in a wider mass range are described for the *0CV*, *1CV LL*, and *1CV DD* topologies. The *2CV*

selection remains as is described in Ref. [10]. Section 2 describes the method by which our signal decays are simulated, Section 3 describes the HLT1 strategy. Section 4 describes the HLT2 strategy, performance, and corresponding implementation. The prospects for the current dataset collected by LHCb in addition to that expected to be collected by the upgraded LHCb detector (during Run 3 of the LHC and beyond) are discussed in Section 5.

## 2  Simulating the signal $B_s^0$ and ALP decays

In order to simulate $B_s^0 \to \gamma\gamma$ decays, $pp$ collisions are generated using PYTHIA [13] with a specific LHCb configuration [15]. Decays of hadronic particles are described by EVTGEN [16], in which final-state radiation is generated using PHOTOS [17]. The interaction of the generated particles with the detector, and its response, are implemented using the GEANT4 toolkit [18] as described in Ref. [20]. This allows accurate simulation of the different topologies of the decay, as photons may interact with the detector itself and subsequently decay to electron-positron pairs.

For the ALP signal, samples are generated using MadGraph v2.6 [21], with parameters taken from the ALP model described in Ref. [22]. The hadronization is performed by PYTHIA, and the rest of the simulation steps are identical to simulating the $B_s^0 \to \gamma\gamma$ decay. Three ALP masses are simulated: $5\,\text{GeV}/c^2$, $10\,\text{GeV}/c^2$, and $15\,\text{GeV}/c^2$.

After the detector response has been simulated, the trigger reconstruction and associated selection requirements are simulated using data taking conditions similar to those of the 2017 LHCb running period, with mean interactions per bunch crossing of 1.3 and center-of-mass-energy of 13 TeV.

## 3  HLT1 strategy

The first trigger software level is required to reduce the input rate by a factor 10 from the output of the hardware trigger level. In order to achieve this, a search is made primarily for high transverse momentum tracks or tracks with a high impact parameter with respect to the primary vertex. A detailed description of the HLT1 selections for 2018 can be found in Ref. [10]: in the case of photon conversions with at least one electron reconstructed as a long track, a generic inclusive single-track, MVA-based trigger selection is used [12]; in all other cases, the HLT1 strategy relies on a custom reconstruction approach using information from the calorimeter collected in the hardware trigger stage. This latter reconstruction technique, imposed by the limited time available for decisions in HLT1, applies simple approximations to calculate the mass of the photon pair with a minimum $E_T$ requirement in a negligible time using only their energies as calculated in L0.

Two selections with a different set of requirements are used: a first one, referred to as HLT1($B$), focused on the selection of $B$ decays, and a second one, not included in Ref. [10], with stricter $E_T$ requirements and with a wider mass range, which extends the reach to ALP masses above the $B$ mass window (HLT1(ALP)). Their requirements are given in Table 1; it is worth noting that the $E_T$ requirements of the HLT1(ALP) are very close to the saturation of the LHCb ECAL when using L0 energy clusters,[1] and therefore the true mass reach is higher than the requirement given in the table. The efficiency of the two selections relative to all candidates passed by the L0 hardware trigger is given in Table 2.

---

[1]Calorimeter clusters in L0 are made up of $2 \times 2$ calorimeter cells. As a consequence, the mass that can be reconstructed at L0 level is limited by the saturation of these 4 cells of the cluster. This limitation will not be present in the final analysis, as the offline reconstruction allows to build larger clusters.

Table 1: Selection applied in the HLT1(B) and HLT1(ALP) triggers. Energies and masses given here are computed with $2 \times 2$ cell clusters.

| Requirement | HLT1($B$) | HLT1(ALP) |
|---|---|---|
| $E_{\mathrm{T}}(\gamma)\,[\mathrm{GeV}]$ | $> 3.5$ | $> 5$ |
| $E_{\mathrm{T}}(\gamma_1) + E_{\mathrm{T}}(\gamma_2)\,[\mathrm{GeV}]$ | $> 8$ | $> 11$ |
| $M(\gamma_1\gamma_2)\,[\mathrm{GeV}/c^2]$ | $[3.5, 6.0]$ | $[6.0, 11.0]$ |
| $p_{\mathrm{T}}(\gamma_1\gamma_2)\,[\mathrm{GeV}/c]$ | $> 2$ | $> 5$ |

Table 2: Percentage efficiency relative to all candidates accepted by the Photon and Electron channels of the L0 hardware trigger for the $B_s^0$ and ALP samples, combining all the $\gamma$ reconstruction modes.

| Efficiency (%) | HLT1($B$) | HLT1(ALP) |
|---|---|---|
| $B_s^0 \to \gamma\gamma$ | $3.3 \pm 0.2$ | - |
| ALP 5 GeV | $6.2 \pm 0.5$ | $0.6 \pm 0.2$ |
| ALP 10 GeV | $5.3 \pm 0.3$ | $6.7 \pm 0.4$ |
| ALP 15 GeV | $3.8 \pm 0.3$ | $10.5 \pm 0.5$ |

## 4 HLT2 strategy

The second trigger software level performs a more complete event reconstruction. In HLT2, over 400 multibody decay signatures are searched for in parallel, with candidate selection information equivalent in quality to that used by analysts offline.

### 4.1 Training sample preparation

The first step in designing an HLT2 strategy is to collect representative samples of signal and background in order to train neural network (NN) classifiers. For the case of the signal, *i.e* $B_s^0$ and ALP decays, simulated data is used, which is generated as described in Sec. 2. In order to describe the background, proton-proton collisions collected by the LHCb collaboration during 2017 are used, in which the high level trigger selected candidates randomly.

Since the intention is to implement a NN inside the trigger software, and in order to generate the largest possible NN training sample, a set of loose requirements are applied to the samples mentioned above. These are inspired by those applied in the analysis of other radiative decays (*e.g.* those in Refs. [23–25]). In summary,

- photons reconstructed as calorimeter energy clusters are required to have an energy above 6 GeV and a transverse energy with respect of the beam direction above 3 GeV;

- photons reconstructed as electron pairs are required to have a transverse momentum in excess of 2 GeV/$c$, have a mass below 60 MeV/$c^2$ and be displaced with respect to the $pp$ collision;

- the sum of transverse momentum of the two photons must be in excess of 6.5, 5.5 and 5 GeV/$c$ for the *0CV*, *1CV* and *2CV* cases, respectively; and

- diphoton candidates are required to have a combined transverse momentum above 3 GeV/$c$ and, in the case of *2CV*, to form a good vertex.

Table 3: Sample sizes for the signal decays and background after reconstruction and trigger requirements.

| Sample | *1CV LL* | *1CV DD* | *0CV* |
|---|---|---|---|
| $B_s^0 \gamma\gamma$ | 9940 | 17368 | 36844 |
| ALPs | 229 | 420 | 233 |
| Background | 228 | 393 | 457 |

In the training of the NN, a random subset of the $B_s^0 \rightarrow \gamma\gamma$ sample is taken, such that efficiencies of the ALP signal (for the 3 different masses under consideration) and $B_s^0$ decay remain similar. The yields of the samples are provided in Table 3.

### 4.2 Multi-layer perceptron training and implementation

A multilayer perceptron (MLP) was chosen for the NN updates. A Scikit-learn [26] implementation was preferred due to the relative simplicity of the topology that allows for quick evaluation in real-time environments in association with the `NNDrone` package [27]. Due to the small training dataset, a LBFGS solver method [28], which shows better results under these conditions, was used. To avoid overtraining a small ($\mathcal{O}(10^{-5})$) regularization parameter was chosen. A simple hidden layer structure was implemented: three neurons in the first and two in the latter. The small dimension was primarily set to avoid large time complexity. Finally, for simplicity, the logistic activation function, which is also used in the output layers, was chosen for the hidden layers.

The model is created as

```
from sklearn.neural_network import MLPClassifier
classifier = MLPClassifier(activation='logistic'
  , alpha=1e-05, batch_size='auto',
  beta_1=0.9, beta_2=0.999, early_stopping=False,
  epsilon=1e-08, hidden_layer_sizes=(3, 2),
  learning_rate='constant', learning_rate_init=0.001,
  max_iter=200, momentum=0.9,
  nesterovs_momentum=True, power_t=0.5, random_state=1,
  shuffle=True, solver='lbfgs', tol=0.0001,
  validation_fraction=0.1, verbose=False, warm_start=False)
```

In order to train the MLP for each topology, different variables are found to have different separation powers. We select those providing significant discrimination between signal and background. The following variables are used as a feature for one or more of the NN models:

- The transverse momentum of the parent $B_s^0$ or ALP candidate (X $p_T$).

- The impact parameter significance of the $B_s^0$ or ALP candidate with respect to the best primary vertex (X IP $\chi^2$), defined as the difference in $\chi^2$ of a given PV reconstructed with and without the considered particle.

- The invariant mass of the electron-positron combination in a photon conversion.

- The probability that the photon candidate is not a $\pi^0$ meson based on a combination of calorimeter information [29] ($\gamma$ prob).

Table 4: Kolmogorov-Smirnov p-values for the comparison of the classifier's distribution between the test and training samples for the different topologies

| Topology | Signal p-value | Background p-value |
|---|---|---|
| 0CV | 0.25 | 0.63 |
| 1CV DD | 0.99 | 0.99 |
| 1CV LL | 0.48 | 0.37 |

- Asymmetry of the $p_{\mathrm{T}}$ of the two photon candidates ($p_{\mathrm{T}}$ asym).

- The ratio of the candidate ECAL energy deposit between the $2 \times 2$ and $3 \times 3$ clusters ($\gamma$ Calo E49).

- The output of a multi-variate classifier [29] trained using various inputs corresponding to calorimeter shape variables ($\gamma$ shower shape).

The signal and background distributions for each topology are given in Appendix A. It should be remarked that, as mentioned above, our classifiers use as inputs the outputs of two other different classifiers, one designed generically to identify calorimeter photons ($\gamma$ shower shape) and one to differentiate photons and $\pi^0$ mesons ($\gamma$ prob). While, ideally, one could integrate the features used in these classifiers into our own one, making use of the ability of algorithms such as Deep Neural Networks as feature extractors, it was decided against this given the high level of complexity in the training of both $\gamma$ shower shape and $\gamma$ prob. The first accounts for the expected deposit shapes in different regions of the calorimeter and excludes the possibility that these are originated by a charged track. The second also relies on the energy value and the calorimeter zone comparing the expected energy deposit of a photon with respect to that of a $\pi^0$. Although both classifiers were trained to maximize its sensitivity for $B$ meson decays, potential extensions of this work could involve integrating all the features into a single classifier to be trained specifically with our signal samples.

The samples are split into training and test data, where half of the data are used for training and the other half used for testing. Optimizations of this splitting, such as the use of techniques like k-folding [30], will be explored in future versions of this algorithm to maximize the size of the training and test samples. The output distributions of the trained models for both the test and training samples are given in Fig. 1. Good agreement can be seen in all training and test comparisons, meaning the models show few signs of overtraining. To quantify this, p-values obtained with the Kolmogorv-Smirnov test [31] can be found in Table 4.

### 4.3 Performance

The performance of each of the models shown in terms of ROC curves is provided in Appendix B. Ultimately, the efficiency of the models depends on the chosen working point. This is driven by resource requirements. The chosen working point of each of the models is shown in Table 5, along with the rejections and efficiencies per sample, per decay topology.

### 4.4 Implementation in the real-time software stack

In order to apply the neural networks in the C++ software stack used to perform event selection in real time, a conversion must take place so that the models trained in the Python implementation are reproduced.

In order to do this, the `NNDrone` framework [27] is used to convert the network weights and bias parameters to `JSON` format. This allows reducing the processing time by an order of magnitude while keeping the same classifier performance. The weights and parameters

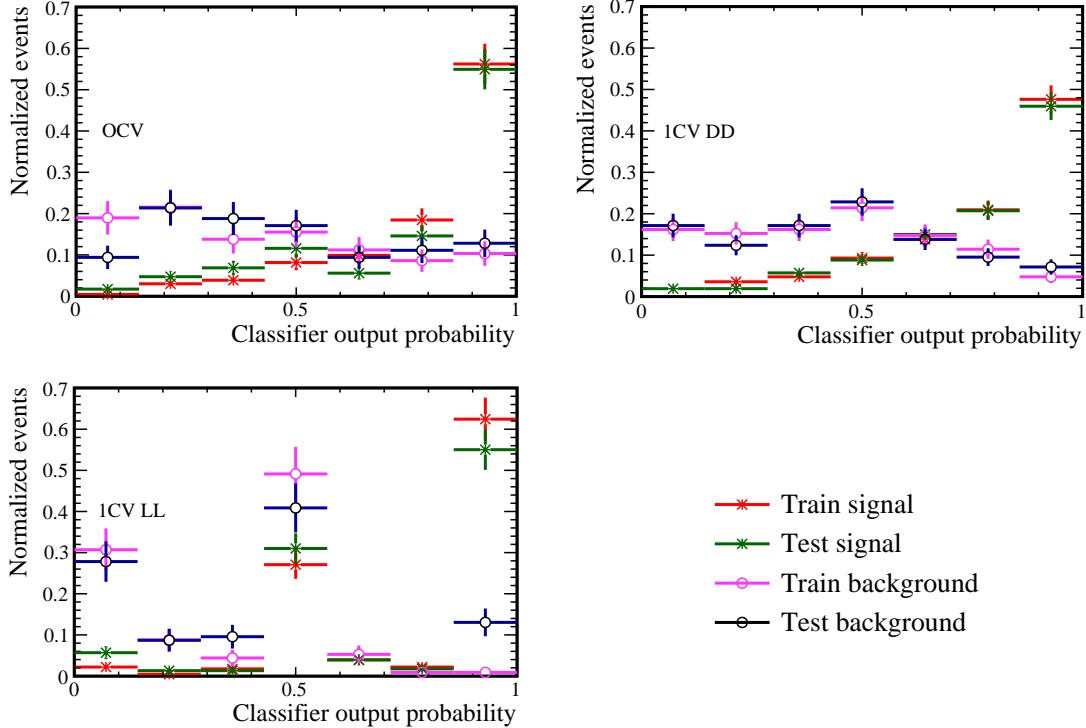

Figure 1: Output probability of the classifier for the different topologies. Upper left: *0CV*, upper right: *1CV DD*, lower *1CV LL*

are then read in by a dedicated tool which uses the JSON parameters to initialise a LWTNN network [32] that reproduces the original model exactly. The reconstructed LWTNN model has been validated to produce identical outputs from the same input values.

# 5 Prospects

As explained in Section 1, the trigger strategy described in this paper is essential to improve the sensitivity of LHCb to $\gamma\gamma$ final states, especially after 2018. In this section, we describe briefly the prospects for the two benchmark analyses studied: the search for the rare decay

Table 5: Percentage efficiency for the $B_s^0$ and ALP samples relative to the reconstructed and loosely selected samples.

| | | 1CV LL | 1CV DD | 0CV |
|---|---|---|---|---|
| Efficiency (%) per channel | $B_s^0 \to \gamma\gamma$ | 67 | 31 | 68 |
| | ALP 5 GeV | 52 | 67 | 71 |
| | ALP 10 GeV | 55 | 50 | 50 |
| | ALP 15 GeV | 62 | 64 | 46 |
| Rejection (%) | | 85 | 90 | 85 |
| MLP requirement | | > 0.70 | > 0.85 | > 0.80 |

$B^0_{(s)} \to \gamma\gamma$ and that of an ALP decaying to a pair of photons. It should be noted that the same final state could also be sensitive to other models, such as those including a composite Higgs together with light scalars [33].

During the full LHCb Run 2 data-taking, selections providing high efficiencies for the reconstruction of $B^0_{(s)} \to \gamma\gamma$ decays were included (see Tab. 2 and Tab. 5). In the same regard, the efficiencies for an ALP with a mass close to that of the $B_{(s)}$ meson is also very high for that period. For the case of an ALP with a mass between $\sim 6$ GeV/$c^2$ and $\sim 12$ GeV/$c^2$, only data collected in 2018 with the strategy described in this document provides significant sensitivity. A similar strategy to that used in 2018 is expected to be used during Run 3 of the LHC, in which LHCb will have been upgraded [34]. Concerning the trigger, the new design [35] including the removal of the first hardware level, should provide similar or higher efficiencies.

The efficiencies reported in this document, together with the fraction of triggered data provided in Ref. [10], are used to roughly estimate the expected sensitivity in both analyses. A $\gamma\gamma$ invariant mass resolution of $\sim 2.5\%$ is assumed for these studies. Additional offline discrimination against the background can be achieved by using similar but more powerful classifiers than those available in HLT2. This additional discrimination is based on the use of larger training samples and variables whose reconstruction is too slow to be performed in real time. Additional background rejections of $\sim 90\%$ can then be achieved with associated signal efficiencies of $\sim 60\%$ for all the photon reconstruction categories included in this document.

For the $B^0_s \to \gamma\gamma$ decay an upper limit $\mathcal{B}(B^0_s \to \gamma\gamma) \lesssim 10^{-5}$ at 90 % CL could be achieved using the Run 2 LHCb dataset. This is around two times the Belle limit, currently the most stringent. Assuming similar efficiencies and backgrounds in Run 3 of the LHC, a simple projection yields $\mathcal{B}(B^0_s \to \gamma\gamma) \lesssim 4 \times 10^{-6}$. This assumption might not hold if the ECAL performance is affected by the larger occupancy expected in Run 3. If a more optimistic background discrimination is assumed (95% background rejection for the same signal efficiency) upper limits of $\mathcal{B}(B^0_s \to \gamma\gamma) \lesssim 6 \times 10^{-6}$ and $\mathcal{B}(B^0_s \to \gamma\gamma) \lesssim 2 \times 10^{-6}$ could be achieved using the Run 2 and Run 3 LHCb datasets, respectively. A discovery, should the SM prediction hold, would probably need to wait until Run 4 or a potential Phase-II LHCb upgrade [36].

Concerning a search for ALPs, our estimations agree with those of Ref. [7]. For the reasons explained previously, the sensitivity with the current dataset is more limited for an ALP with a mass in the $\sim 6 - 12$ GeV/$c^2$ range. In the most sensitive region, decay constants below $\sim 0.3$ TeV could be excluded using the LHCb Run 2 dataset. Keeping the same efficiency as in 2018 for Run 3 would provide an increase to $\sim 0.4$ TeV for the $\sim 4 - 12$ GeV/$c^2$ mass range. No other experiment is expected to contribute to the measurement in this mass range in near future.

## 6 Summary

Di-photon selections were first implemented in the LHCb trigger in 2015 focusing on the $B^0_s$ decay. In this paper, we have detailed the selection modifications required to expand the search region to allow sensitivity to undiscovered ALPs in the 2018 trigger. In order to do this and remain within resource budgets, neural network models have been introduced using the NNDrone framework to ensure fast evaluation. This is the first time such models have been used to directly select multibody candidates in real time.

# A  Neural network feature distributions

The signal and background distributions for each topology are shown in Figs. 2, 3, and 4, respectively, where "signal" refers to the combination of all signal modes (i.e., $B_s^0 \to \gamma\gamma$ and the ALP at the three mass points of reference).

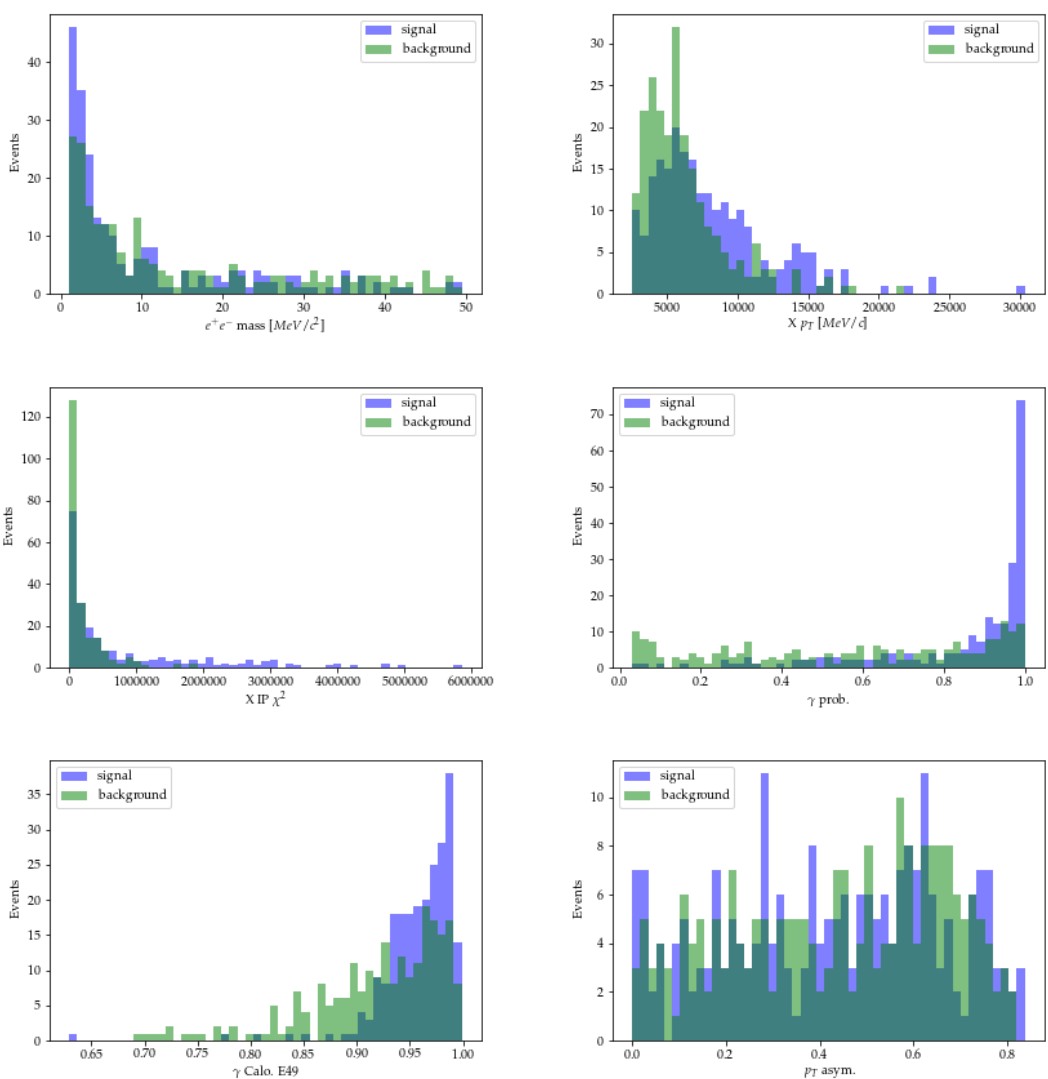

Figure 2: Signal and background distributions of information used to train the *1CV LL* classifier. Variables are explained in the text.

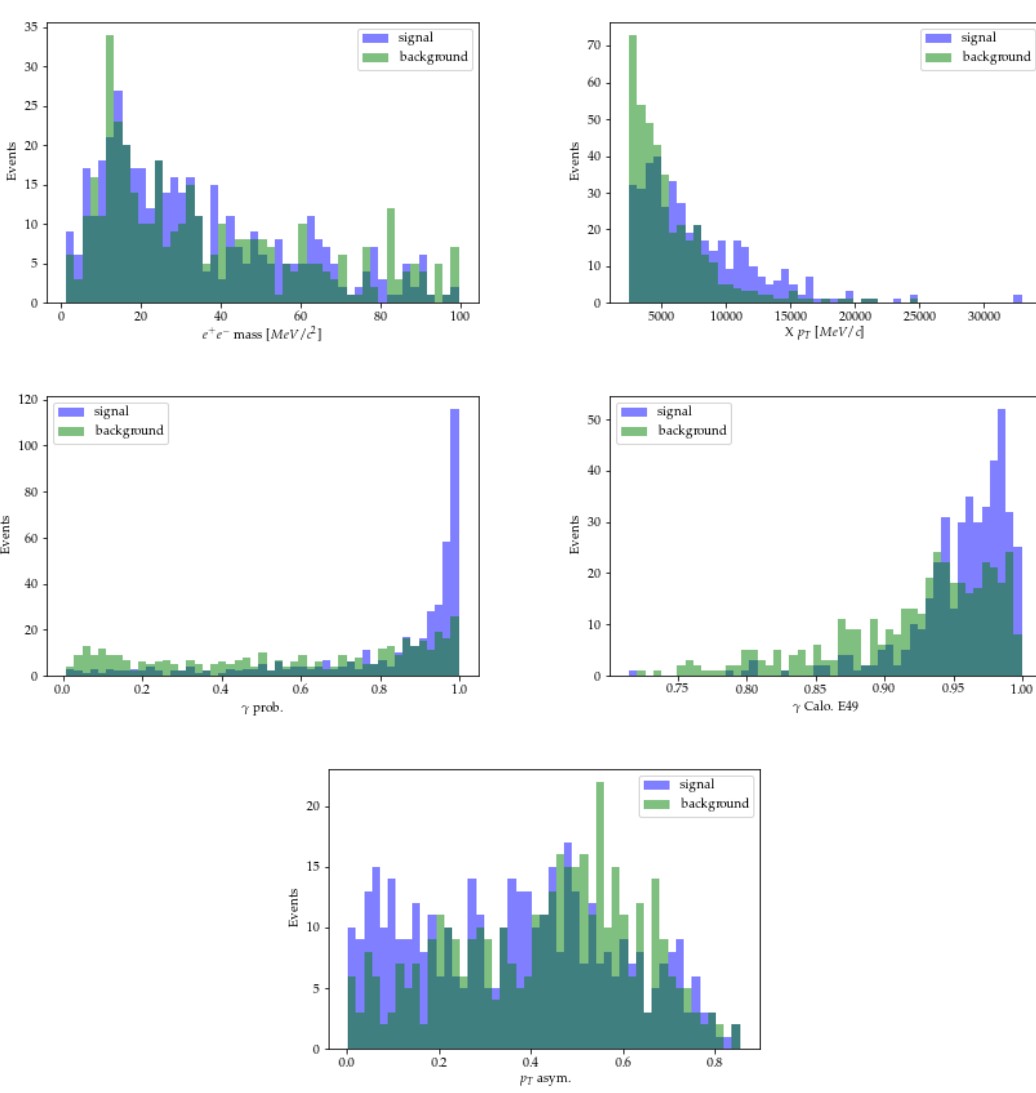

Figure 3: Signal and background distributions of information used to train the *1CV DD* classifier. Variables are explained in the text.

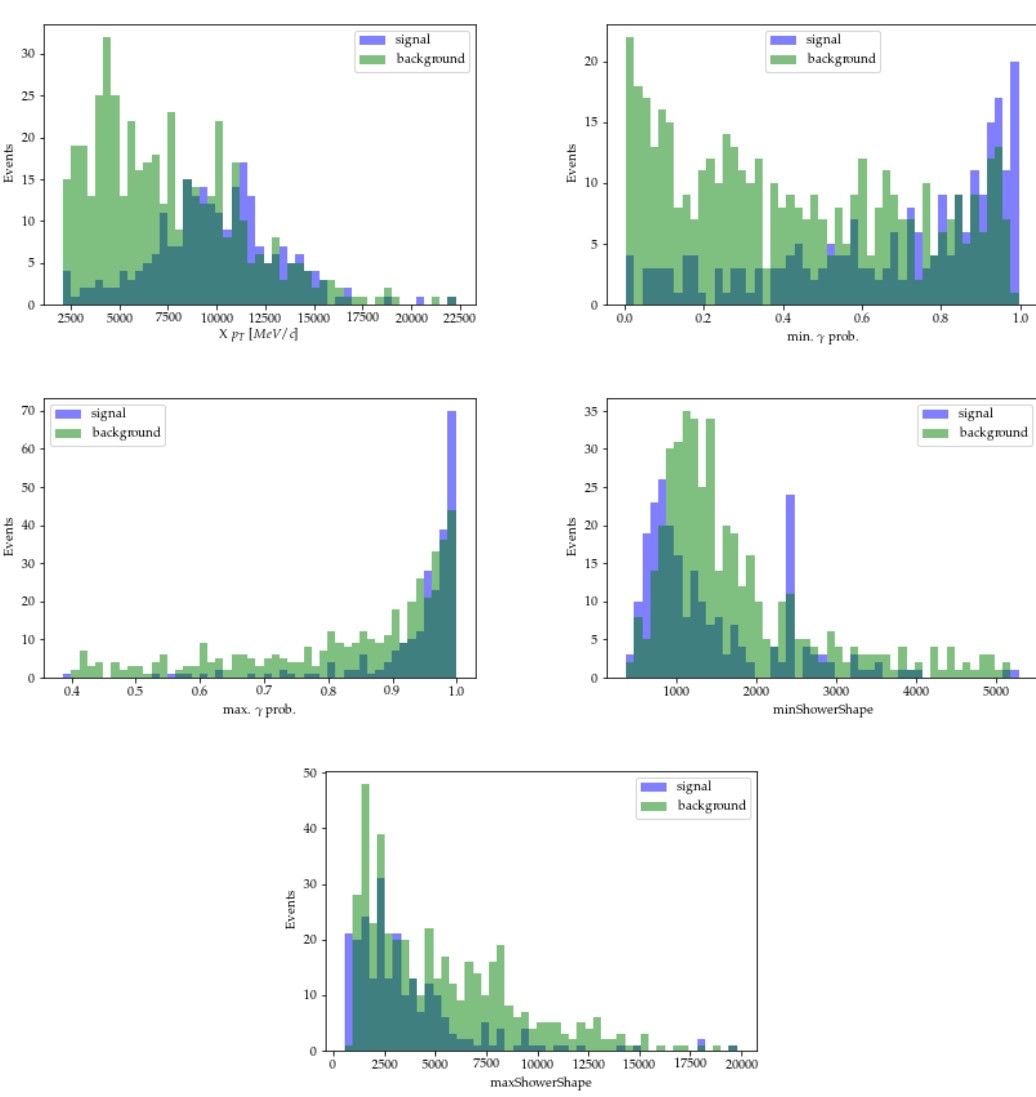

Figure 4: Signal and background distributions of information used to train the *OCV* classifier. Variables are explained in the text.

# B  ROC curve performances of the NN models

The performance of each of the models is shown in terms of ROC curves in Fig. 5, which display the signal efficiency versus background rejection power.

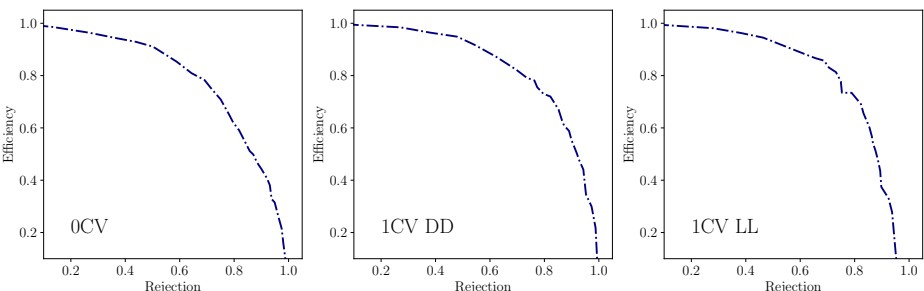

Figure 5: ROC curves for the test data using the different topologies. 0CV NN (left), 1CV DD NN (center),1CV LL NN (right) .

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
