# Peer review of "Real-time discrimination of photon pairs using machine learning at the LHC"

_SciPost Physics, doi:SciPost Phys. 7, 062 (2019)_

## Round 1 · Referee Report · Anonymous (Referee 1) · 2019-8-6

Strengths

  1. The paper gives a clear and succinct account of the trigger strategy at LHCb, for a particularly challenging topology.
  2. The paper demonstrates a successful and novel application of a neural network in a real-time environment. Moreover, the application to searches for axion-like particles is of particular relevance given the popularity of these as candidates for dark matter.

Weaknesses

  1. The paper is lacking in detail regarding some of the choices for designing the network and regarding its performance.

Report

This publication details the strategy adopted at the LHCb experiment for triggering events with photon pairs consistent with B-hadron decays. The strategy employs the use of a neural network, trained using simulated events, to efficiently select such decays in real-time. This trigger can also be exploited to search for hypothetical axion-like particles. This trigger strategy is clearly a vital component of analyses targeting diphoton decays and the use of neural networks to provide good separation of the signal and background under timing constraints is novel. While I certainly recommend publication of this paper, I have a few suggestions for the authors to consider below.

While the paper clearly describes the strategy for training the neural network, including appropriate details on the choice of features and the simulation of signal events, there is limited discussion regarding the optimisation of the network and its performance, which would be of interest to the reader.

In section 2, page 2, paragraphs 1 and 2 describe the event generation for the $B_{s}^{0}$ and ALP signals which are different for the two. It would have been useful to know if the trigger performance would be sensitive to the choice of generators and whether other generators were investigated, or whether such an investigation would not be possible.

In section 4.2, page 4, paragraph 1, the MLP training parameters used are listed. However, there is no supporting discussion which motivates these choices. Were these tuned in some way or are these parameters known to be optimal for this particular problem. It would be of interest to the reader to know these details as other applications may find similar choices for the training parameters to be optimal.

In section 4.2, page 5, paragraph 2, it is stated that "Good agreement" can be seen for the training and testing samples. However, the authors do not support this with any metric, such as a KS test or other 2-sample goodness of fit test. It would be a much stronger claim with a supporting statistical test such as this.

Section 4.3 describes the performance of the classifier in terms of the rejection of the background and signal efficiencies. However, there is no mention of the performance in terms of the processing time (although it is hinted at in Section 4.4). Given the application in this case, the need to maintain a low processing time is a crucial aspect and often not considered in other applications of neural-networks. I would find it very interesting to know if the authors have studied how the design of the network could be optimised with respect to the choice of training parameters, especially if it was already optimised in this case. It would also help to reinforce the statement in Section 6 that the NNDrone framework ensures fast evaluation.

Section 5, page 6, paragraph 3 describes an estimate for the sensitivity achievable for the branching fraction limit of $B_{s}^{0}\rightarrow \gamma\gamma$. It would have been interesting to see this analysis in more detail, instead of just stating that the full analysis would use larger data samples for training and more complicated variables. Was this study documented elsewhere which could be cited?

Requested changes

  1. Section 1, page 1, paragraph 2. "... sets indirectly constraints to the New Physics ... " should read "... sets indirect constraints on the New Physics ...". In the same paragraph, I find "used for their discovery" a bit vague. Rather the authors could write that the coupling to photons allows for detection of ALPs.

  2. Section 4.1, page 4, paragraph 3. Is this efficiency with respect to all 3 mass points combined or an average efficiency for the 3 mass points - please clarify this point.

  3. Figure 1. The figures are rather small here. I would ask the authors to increase the label sizes for better visibility. It is also more appropriate in the caption to refer to the figures with (left), (centre) and (right) or at least use the same labelling in the caption as the figures (eg "No conversions vs 0CV NN)

  4. Section 4.2 - See main report for discussion of choices of the training parameters.

  5. Section 4.2, page 5, paragraph 2 - Add statistical test for comparison of training and test samples (see main report)

  6. Section 4.3 - Add discussion of the performance in terms of processing time and whether an optimisation in this respect was studied (see main report)

  7. Table 4. The MLP requirement is presumably a lower bound on the "classifier" quantity from Figure 1? Please clarify this in the table or in the caption. While the expert reader can probably assume this, the general reader might not.

  8. Section 5, page 6, paragraph 3 - See main report for adding details on the sensitivity estimate

  9. Appendix A. Not clear what "combination" of all signal models. -> see request #2.

  • validity: high
  • significance: good
  • originality: good
  • clarity: high
  • formatting: good
  • grammar: good

Author:  Xabier Cid Vidal  on 2019-10-02  [id 616]

(in reply to Report 1 on 2019-08-06)
Category:
answer to question

Dear referee,

thanks a lot for your inputs to our manuscript. Your suggestions and questions have certainly improved our manuscript.
Please find attached replies to your questions/comments. We hope you are satisfied with them and with the new version of the paper.
Kind regards,

Xabier (for the authors)

Attachment:

replies_ref1.pdf

Anonymous on 2019-10-24  [id 631]

(in reply to Xabier Cid Vidal on 2019-10-02 [id 616])
Category:
remark
objection

Dear Authors,

Thank you for considering my suggestions and for your reply. I certainly think the draft has been improved, but I have one follow-up to the responses.

  1. While Fig 1. has improved and is now very readable, I find the figures in the Appencides still very hard to read. I realise it may be difficult to regenerate these figures but can the labels be improved (by editing the pdf files or similar) at least? As they are, the figures just really are not publication quality.

Kind regards, Your referee

---

## Round 1 · Referee Report · Anonymous (Referee 2) · 2019-8-28

Strengths

1- Application oriented, not just a proof of principle on some simplified toy dataset 2- Successfully faces the challenge of a real-time implementation 3- Production-ready algorithm

Weaknesses

1- The study seems to lack full quality of a scientific publication and, in a few parts, reminds me more of a collaboration-internal note 2- In a few points (see report) I have the impression that the chosen strategy is not the most effective 3- There are many points in which arbitrary choices are not motivated

Report

The paper discusses a real-life application of Machine Learning in an LHC experiment. Unlike many existing proof-of-concepts studies, this work addresses a specific problem in a realistic environment (e.g., with computational and latency constraints) and with a realistic dataset. In this respect, the topic is certainly of interest for a publication. On the other hand, I have the impression that the work and its description are don't meet yet the standards of a scientific publication. I think that the authors should take the time to improve the draft and submit a revised version, that could then be considered for publication. I list below a few suggestions along this line (see "Requested Changes")

Requested changes

A few questions 1- why at the beginning of the introduction Bs and not Bd is mentioned? 2- While I agree that the QCD axion would couple to the gluon, is this a generic feature of anything that people would call ALP? 3- It is not clear to me which L0 selection is implied here and which effects it has on the downstream analysis. This is barely mentioned at some point, but more details should be given. 4- The choice of giving a cryptic code snippet to describe the network architecture is quite low level and dumps on the reader the task of retrieving the meaning of any parameter from the scikit learn manual. Under the assumption that the results don't depend on the library used for the implementation (I don't doubt this), it is preferable to have a text-based description of the architecture, the hyper-parameter setting, the training procedure, etc. 5- the choice of input features is very arbitrary and leaves a lot of open questions. Given the demonstrated ability of Deep Neural Networks as feature extractors, what is the point of using as input the score of another ML algorithm instead of the inputs used to train those algorithms? 6- Fig.1 should be remade using histogram line rather than dots. In these plots and in those shown in appendix, one would need to use more statistics and/or less bins to reduce the effect of random statistical fluctuations 7- In Section 4.4, I would have expected some discussion about the latency, memory footprint, and in general some quantification of the performance. 8- A plot should be added to show what is said at the end of Section 5

A few editorial remarks 1-In the abstracts, ALP is not a b-decay. An ALP-mediated decay to a diphoton final state would be a decay 2- In the abstract, The fast -> A fast 3- In the introduction (first sentence) final state of a B meson 4- In the introduction, Standard Model of particle physics 5- In the introduction, L0 is undefined 6- In section 4.1, data is plural 7- In Table 4, the first column is not the efficiency. It is the process. The other three columns refer to the efficiency. 8- In section 5, Sect. -> Section

  • validity: good
  • significance: good
  • originality: ok
  • clarity: low
  • formatting: below threshold
  • grammar: acceptable

Author:  Xabier Cid Vidal  on 2019-10-02  [id 615]

(in reply to Report 2 on 2019-08-28)
Category:
answer to question

Dear referee,

thanks a lot for your careful reading of our manuscript. Your suggestions and questions have certainly improved to improve our manuscript.
Please find attached replies to your questions/comments. We hope you are satisfied with them and with the new version of the paper.
Kind regards,

Xabier (for the authors)

Attachment:

replies_ref2.pdf

---

## Round 2 · Referee Report · Anonymous (Referee 2) · 2019-10-5

Report

I reviewed v2 of the paper and I think that, given the modifications made, the paper is to a sufficient quality level to be accepted for publication.

Only one of the point was not addressed (remaking the plots in appendix with more statistics), due to a major difficulty (person in charge left the field). Considering that the plots in question are in the appendix and not in the main body, I have the impression that this should not be seen as a showstopper.

I am still not 100% satisfied with the justification behind some choice. The architecture chosen for the MLP is quite obsolete under several aspects (e.g., the choice of the activation function). The authors say that the choice was done for simplicity, but there is nothing complicated in using, for instance, a more state-of-the-art function like a ReLU. This doesn't mean that what was done is wrong. Still, it leaves the impression that this work was put together in a hurry (partially confirmed by the authors' replies).

All in all, I am left with the impression that things could have been done better, but that what was done is correct and relevant enough to be published.

---

## Round 2 · Referee Report · Anonymous (Referee 2) · 2019-10-5

Report

I reviewed v2 of the paper and I think that, given the modifications made, the paper is to a sufficient quality level to be accepted for publication.

Only one of the point was not addressed (remaking the plots in appendix with more statistics), due to a major difficulty (person in charge left the field). Considering that the plots in question are in the appendix and not in the main body, I have the impression that this should not be seen as a showstopper.

I am still not 100% satisfied with the justification behind some choice. The architecture chosen for the MLP is quite obsolete under several aspects (e.g., the choice of the activation function). The authors say that the choice was done for simplicity, but there is nothing complicated in using, for instance, a more state-of-the-art function like a ReLU. This doesn't mean that what was done is wrong. Still, it leaves the impression that this work was put together in a hurry (partially confirmed by the authors' replies).

All in all, I am left with the impression that things could have been done better, but that what was done is correct and relevant enough to be published.

---

## Round 2 · Author Response

Dear editor,

Thanks for considering our manuscript for publication. We have done our best to address the comments kindly provided by the referees. We believe they have helped significantly to improve the quality of the manuscript.
Kind regards,

Xabier (for the authors)

---

## Round 2 · List of Changes

• Improved description of trigger selection steps and selection cuts
  • Included discussion concerning the choice of input features of our classifiers
  • New discussion about the parameters chosen in the training
  • Improved quality of the plots and added more information concerning the similarity of test and training samples
  • Different editorial fixes

---

## Editorial Decision

published